# Phloroglucinol-Based Carbon Quantum Dots/Polyurethane Composite Films: How Structure of Carbon Quantum Dots Affects Antibacterial and Antibiofouling Efficiency of Composite Films

**DOI:** 10.3390/polym16121646

**Published:** 2024-06-11

**Authors:** Zoran M. Marković, Dušan D. Milivojević, Janez Kovač, Biljana M. Todorović Marković

**Affiliations:** 1Vinča Institute of Nuclear Sciences, National Institute of the Republic of Serbia, University of Belgrade, 11158 Belgrade, Serbia; dusanm@vinca.rs; 2Department of Surface Engineering, Jozef Stefan Institute, Jamova 39, SI-1000 Ljubljana, Slovenia; janez.kovac@ijs.si

**Keywords:** phloroglucinol, carbon quantum dots, solvothermal procedure, composite films, nano-electrical microscopy, nano-mechanical microscopy, antimicrobial, singlet oxygen

## Abstract

Nowadays, bacteria resistance to many antibiotics is a huge problem, especially in clinics and other parts of the healthcare system. This critical health issue requires a dynamic approach to produce new types of antibacterial coatings to combat various pathogen microbes. In this research, we prepared a new type of carbon quantum dots based on phloroglucinol using the bottom-up method. Polyurethane composite films were produced using the swell–encapsulation–shrink method. Detailed electrostatic force and viscoelastic microscopy of carbon quantum dots revealed inhomogeneous structure characterized by electron-rich/soft and electron-poor/hard regions. The uncommon photoluminescence spectrum of carbon quantum dots core had a multipeak structure. Several tests confirmed that carbon quantum dots and composite films produced singlet oxygen. Antibacterial and antibiofouling efficiency of composite films was tested on eight bacteria strains and three bacteria biofilms.

## 1. Introduction

Nosocomial infections take place in healthcare facilities including clinics, long-term facilities, and ambulance settings. Numerous clinic patients are affected by infections during hospitalization (3.2% in the USA, 6.5% in the EU, and much higher numbers in the undeveloped countries) [1,2]. The pathogens that are usually involved in hospital-acquired infections include *Streptococcus* spp., *Acinetobacter* spp., *enterococci*, *Pseudomonas aeruginosa*, *Coagulase-negative staphylococci*, *Staphylococcus aureus*, *Bacillus cereus*, *Legionella* and *Enterobacteriaceae* family members, namely, *Proteus mirabilis*, *Klebsiella pneumonia*, *Escherichia coli*, and *Serratia marcescens*. All these microbes can be transmitted through various pathways: person to person, polluted water, infected individuals, infected healthcare personnel skin, or via inanimate surfaces [3]. Apart from the pathogens mentioned above, multidrug-resistant microbes include methicillin-resistant *Staphylococcus aureus*, vancomycin-resistant *enterococci*, *Pseudomonas aeruginosa*, and *Klebsiella pneumonia*.

This critical health issue requires a dynamic approach to produce antimicrobial surfaces to cut the transmission path of microbes from various highly touched objects such as doors, mobile phones, and currency to clinical personnel and patients [4]. The key parameters which affect the efficacy of antimicrobial surfaces are their wettability, surface roughness, stiffness, surface texture, surface chemistry of used antimicrobial agent, and the charge transfer between pathogens and the surface [5]. One of the strategies for pathogen eradication is antimicrobial photodynamic therapy, which is defined as a noninvasive therapeutic modality for the prevention of different infectious diseases [6]. It is defined as an O_2_-dependent photochemical reaction that takes place under visible light irradiation of material called a photosensitizer, leading to generation of reactive oxygen species [7].

Physicochemical characterization and antimicrobial effects of carbon quantum dot/polyurethane composite films are described in this paper. Carbon-based dots include graphene quantum dots, carbon quantum dots, carbon nanodots, and carbon polymerized dots [8]. This material has unique structural, chemical, optical, and biomedical properties, including the utilization as antibacterial or anticancer agents [9,10,11,12,13].

Carbon quantum dots were prepared from phloroglucinol by the solvothermal method. Phloroglucinol (1,3,5-trihydroxybenzene) is a highly symmetrical organic molecule [14]. It is used as a reactant for the synthesis of a variety of industrial products: pharmaceuticals, dyes, and explosives [15,16]. This molecule exhibits a broad spectrum of pharmacological activities: antibacterial, antiviral, antifungal, antioxidant, and antidepressant [17,18,19]. Carbon quantum dots can be produced by condensation of phloroglucinol in liquid (ethanol, 1,2-pentanediol) or air at elevated temperatures [20,21,22,23,24].

These dots have plenty of hydroxyl groups on their basal plane as well as carbonyl groups on the edges of honeycomb structure [25]. Gohda et al. estimated the structure of phloroglucinol-based CQDs which represents electron rich sp^2^ islands of various sizes interconnected with C-O-C bonds [24]. By using atomic force microscopy operated in tapping mode, electrostatic, and viscoelastic modes, we determined their morphology, electronic properties, and Young’s modulus of elasticity, which proves that the structure was correctly estimated. We performed antibacterial tests of these composite films against eight bacteria strains and three bacteria biofilms. The cytotoxicity of these composite films was investigated on HaCat cells.

## 2. Materials and Methods

### 2.1. Materials

Phloroglucinol (Carl Roth 97%, Karlsruhe, Germany), acetone (Honeywell p.a., Charlotte, NC, USA), white nylon membrane filter 100 nm pore size (Tisch Scientific, Cleves, OH, USA), Singlet Oxygen Sensor Green (Invitrogen, Thermo Fischer Scientific, Waltham, MA, USA), and methanol (Merck, Rahway, NJ, USA), D-Acetone (Sigma Aldrich, Rahway, NJ, USA) were purchased and used as received. Super-clear medical-grade high-stretch polyurethane waterproof transparent TPU film was purchased by DG Xionglin New Materials Technology, China. The polymer film thickness was 0.2 mm.

### 2.2. Synthesis of Carbon Quantum Dots and Polyurethane Composite Films

Carbon quantum dots (PHL-CQDs) were prepared by one-step solvothermal procedure [26,27]. A total of 1 g of phloroglucinol and 2 mL of concentrated H_2_SO_4_ were mixed in 100 mL ethanol. Then, clear solution was treated in the PTFE-lined autoclave for 12 h at 180 °C. A dark red product was produced. It was thoroughly washed, dried in air, redissolved in acetone, filtered, and centrifuged at 4000 rpm. The supernatant was collected and used for experiment.

The PHL-CQDs/polyurethane (PHL-CQDs/PU) composite films were prepared as follows: pieces of polyurethane samples (25 × 25 × 1 mm^3^) were dipped in PHL-CQDs solution in acetone (50 mL). The concentration of the PHL-CQDs was 98 mg/mL. The swelling–encapsulation–shrink method was used to encapsulate the PHL-CQDs in PU [28]. The swelling procedure lasted 1 min at room temperature. The PHL-CQDs composite films were dried at 80 °C for 12 h in a vacuum furnace to eliminate acetone from the composite films.

### 2.3. Sample Characterization

The surface morphology of PHL-CQDs nanoparticles and PHL-CQDs/PU composite films was visualized by atomic force microscopy (AFM) with an AC160 cantilever (MFP-3D Origin, Asylum Research, Oxford Instruments, Santa Barbara, CA, USA). To determine particle size (diameter and height), the microscope was operated in tapping and viscoelastic mode at room temperature [29]. PHL-CQDs nanoparticles’ diameter and height were calculated from more than 20 images in Gwyddion software version 2.64 [30]. Nano-mechanical measurements were performed in amplitude modulation–frequency modulation viscoelastic mapping mode (AC160 cantilever which operated at first and second harmonic) [31]. Frequency, amplitude, and phase of the two modes with a contact mechanics model were used to determine distribution of Young’s modulus.

Nano-electrical microscopy (electrostatic force; EFM) measurements were performed at room temperature with an MFP 3D Origin (Asylum Research, Oxford Instruments, Santa Barbara, CA, USA). A Si cantilever coated with Ti-Ir (ASYELEC-01) was used. A cantilever resonant frequency ν_0_ of 75 KHz, Q factor of 156 and a spring constant k of 2.8 N/m were measured using a thermal noise method. Tip voltage of 3 V and drive amplitude of 50 mV was typically used. Nano-electrical microscopy was performed on the surface of PHL-CQDs nanoparticles deposited on freshly cleaved mica by spin-coating method and PHL-CQDs/PU composite films. Mica and polymer composite films were attached to AFM/SPM Stainless Steel Metal Specimen Support Disc, Ø15 mm (Ted Pella, Redding, CA, USA) with Loctite Superglue. The distribution of charge carriers in the PHL-CQDs nanoparticles in the PU was studied by nano-electrical microscopy as well.

Surface X-ray photoelectron spectroscopy (XPS) analysis of PHL-CQD nanoparticles and PHL-CQD/PU composite film samples was performed on a PHI-TFA XPS spectrometer manufactured by Physical Electronics Inc. equipped with an Al monochromatic source and a hemispherical electron energy analyzer. The diameter of the analysis area was 0.4 mm and the analysis depth was 3–5 nm. During the XPS analyses, the transmission energy was 29 eV, while the energy resolution was 0.65 eV. Two spots were analyzed and the XPS spectra were highly reproducible. The C1s spectrum received a binding energy of 284.8 eV, which is characteristic of C-C/C-H bonds. Quantification of surface composition was performed based on XPS peak intensities, taking into account the relative sensitivity factors provided by the instrument manufacturer [32]. XPS data were processed with Multipak version 9.9.

Fourier transform infrared spectroscopy (FTIR) measurements of PHL-CQDs nanoparticles and PHL-CQDs/PU composite films were performed on a Nicolet iN10 Thermo Fischer Scientific operated in the ATR mode. PHL-CQDs nanoparticles samples were deposited on Si substrates by drop-casting method. Spectra were recorded in the range of 400 to 4000 cm^−1^ at room temperature. Spectral resolution was 4 cm^−1^.

Optical properties of colloidal PHL-CQDs nanoparticles and PHL-CQDs/PU composite films were investigated by UV–Vis spectrometry (Unispec2 LLG spectrophotometer, Meckenheim, Germany) and photoluminescence spectroscopy (Fluoromax+4 spectrofluorometer, Horiba, Kyoto, Japan). The range used for absorption measurements was from 200 to 700 nm at room temperature.

### 2.4. Production of Reactive Oxygen Species

Singlet oxygen production of PHL-CQDs/PU composite film samples was measured using Sensor Green Singlet Oxygen (SOSG) as fluorescence probe. Direct measurement of singlet oxygen production of PHL-CQD was accomplished by measuring the luminescence of ^1^O_2_ at 1270 nm [33].

The procedure for the first method is the following: The concentration of SOSG in methanol solution was 12 µmol. The sample (8 × 8 mm^2^) were dipped in methanol solution of SOSG. PHL-CQDs/PU composite films were irradiated by blue lamp (3 W, V-TAC, Bulgaria) at a wavelength of 470 nm. The sample irradiation time was in the range from 5 to 120 min, and immediately after irradiation, the measurements were conducted on a Fluoromax+4 spectrofluorometer (Horiba, Kyoto, Japan). The PL spectra were measured under 488 nm excitation in the range of 500 to 800 nm. The measurement step was 1 nm.

The procedure for the second method is the following [34]: Emission spectra of singlet oxygen at 1270 nm were obtained on a FluoroMax+4 fluorimeter (Horiba, Kyoto, Japan), with 900 grooves/mm emission monochromator gratings blazed at 1500 nm for NIR range. An ozone-free xenon lamp of 150 W was used as the light excitation source. The emission spectra were measured in the range 1200 to 1300 nm using a thermoelectrically cooled InGaAs detector. Absorption of PHL-CQDs colloid in deuterated acetone was adjusted to be 0.1 at 470 nm. Quartz cuvettes were used for measurement.

### 2.5. Antibacterial Activity

The International standard ISO 22196:2011 (Plastics—Measurement of antibacterial activity on plastic surfaces) was used to determine antibacterial activity of PHL-CQDs/PU composite films [35]. The following species of bacteria were used: *Staphylococcus aureus* NCTC 6571 (*S. aureus*), *S. aureus* MRSA ATCC 43300 (MRSA), *Enterococcus faecalis* ATCC 29212 (*E. faecalis*), *Pseudomonas aeruginosa* ATCC 10332 (*P. aeruginosa*), *Klebsiella pneumonie* ATCC BAA2146 (*K. pneumonie*), *Listeria monocytogenes* NCTC 11994 (*L. monocytogenes*), *Escherichia coli* NCTC 9001 (*E. coli*), and *Acinetobacter baumanii* ATCC 19606 (*A. baumanii*). Sample testing was conducted in triplicate. Control and composite films samples were cut as 25 mm × 25 mm square samples. Before testing, all samples were washed by ethanol. As control, we used nonirradiated samples. Other samples were incubated under blue light during 1 h. All samples were inoculated with 0.2 mL bacteria suspension of 1–5 × 10^8^ cell/mL. The investigated inoculums were coated with films (24 mm × 24 mm), and Petri dishes with samples were incubated for 24 h at 36 °C, humidity 90%. Bacteria were recovered with 5 mL of SCDLP, and 10 µL (different dilutions) were plated on LB agar and incubated for 40 to 48 h, after which bacterial colonies were counted. A blue lamp was placed 50 cm from the samples to obtain homogeneous samples irradiation. Calculation of antibacterial activity was presented in our previous research [27].

### 2.6. Antibiofouling Activity Testing

Antibiofouling activity of PHL-CQDs/PU composite films was tested on the following bacteria strains: *Staphylococcus aureus* NCTC 6571, *Pseudomonas aeruginosa* ATCC 10332, and *Escherichia coli* NCTC 9001. Composite films were gently wiped with 70% ethanol, dried, and irradiated by the blue light during 1 h (except nonirradiated control). Then, the composite films were placed in the wells of 6-well microtiter plates using sterile forceps. Overnight cultures of different bacteria were diluted 1000-fold (1–5 × 10^8^ cell/mL) and added to 5 mL of fresh, sterile LB medium. In each well, fresh LB medium was added (total volume of 5 mL covering the materials). A total of 5 µL of overnight cultures were added in test wells. To control wells, 5 µL of fresh LB medium were added. Plate incubation was conducted for the last 24 h at 37 °C, including rotation with 110 RPM.

The LB medium was removed after incubation. The PHL-CQDs/PU composite films were washed two times with dH_2_O and then dried for 20 min. A 0.1% crystal violet solution was added to each well in a total volume of 3 mL, and composite films were incubated with the stain for 20 min. Then, the PHL-CQDs/PU composite films were washed twice with dH_2_O and dried. After 20 min, a solution of 30% acetic acid was added to each well in a total volume of 3 mL. After 100 µL of acetic acid from each well was transferred to wells of a 96-well microtiter plate, absorbance spectra at 550 nm were measured. The PHL-CQDs/PU composite films with bacteria were incubated in triplicate. Control samples were incubated in duplicate.

### 2.7. Cytotoxicity

To investigate cytotoxicity of investigated composite films, we used standard MTT assay and methods adequate for composite films testing (ISO Standard 10993-5:2009) [36,37]. Testing was conducted in triplicate. Human epidermal keratinocyte line (HaCaT) cell line (T0020001) was commercially obtained from AddexBio Technologies (San Diego, CA, USA). These are in vitro spontaneously transformed keratinocytes from histologically normal skin. All other details related the applied procedure can be found in ref. [27].

## 3. Results

### 3.1. Surface Morphology of PHL-CQDs Nanoparticles

To investigate the surface morphology of PHL-CQDs nanoparticles, we used AFM operated in tapping mode (height retrace) and viscoelastic mode (Young’s modulus of elasticity)—Figure 1a–d. Figure 1a shows that PHL-CQDs nanoparticles have quasi-spherical shape, whereas the mean size of these dots is around 6.2 nm—Figure 1c. The statistical calculation of particle size distribution was performed based on the diameter of more than 1000 particles observed by software (Asylum Research Software, version 14). The mean height of these dots is approximately 0.15 nm (inset of Figure 1c).

Reference value of Young’s modulus of elasticity of mica E_mica_ is in the range 140 to 210 GPa [38]. The Young’s modulus was from 2 to 7 GPa lower than E_mica_ for dots with diameters up to 5 nm, whereas for dots with higher diameters (more than 13 nm), Young’s modulus was from 10 to 12.5 GPa lower than E_mica_ (Figure 1c,d). This result indicates that larger PHL-CQDs nanoparticles are considerably softer than smaller. Under small nanoparticles, we considered particles with diameters up to 5 nm. Under big nanoparticles, we considered particles with diameters higher than 5 nm.

### 3.2. Nano-Electrical and Nano-Mechanical Properties of PHL-CQDs Nanoparticles and PHL-CQDs/PU Composite Films

Electrostatic force microscopy enables the determination of electrical properties of certain nanoparticles or the surface of polymer composites, i.e., determination of surface potential or even charge distribution on the sample surface [39].

Figure 2a,b present the AFM image and corresponding EFM phase image of PHL-CQDs nanoparticles. Figure 2c,d show surface profiles of EFM phase signal of small (number 1) and big (number 2) PHL-CQDs nanoparticles. From this figure, we could observe that small PHL-CQDs nanoparticles have only electron-rich regions (black EFM phase signal), whereas big PHL-CQDs have electron-rich and electron-poor regions (black and yellow EFM phase signal, respectively).

In this way, electrostatic force microscopy reveals that small PHL-CQDs nanoparticles have conductive regions with sp^2^ hybridization only, whereas big nanoparticles have mixed conductive and nonconductive regions with sp^2^ and sp^3^ hybridization (small sp^2^ islands in sp^3^ matrix).

In addition to EFM and viscoelastic microscopy of PHL-CQDs nanoparticles, we also studied the electrical and nano-mechanical properties of PHL-CQDs/PU composite films. Figure 3a,b show EFM and Figure 3c,d show viscoelastic microscopy of PHL-CQDs/PU composite films. In Figure 3a,b, we can identify CQDs by detection of electrons (black voids inside red circles in Figure 3b) in the interior of the polyurethane matrix. The dots are distributed inside the polymer in the form of clusters. The root-mean-square (RMS) roughness of composite films is 9.77 nm, compared to 3.14 nm of neat polyurethane films. After CQDs encapsulation, RMS of composite films increased three times. Figure 3c,d show viscoelastic properties of PHL-CQDs/PU composite films (height retrace and Young’s modulus retrace). It was figured out that the average Young’s modulus was 2.51 GPa. CQDs with considerably higher value of Young’s modulus could not be detected on the surface of the polymer composite.

### 3.3. Chemical Composition PHL-CQDs Nanoparticles and PHL-CQDs/PU Composite Films

XPS and FTIR measurements were conducted to analyze chemical composition of both PHL-CQDs nanoparticles and PHL-CQDs/PU composite films samples. XPS survey spectrum from the surface of the PHL-CQDs nanoparticles is presented in Figure 4a. Table 1 lists the elements detected in the PHL-CQDs nanoparticles sample. It can be observed that the main elements are carbon (81.9 At%), oxygen (16.8 At%), and nitrogen (1.3 At%). The presence of nitrogen in the structure of PHL-CQDs nanoparticles can be explained by the effects of environments.

The carbon C 1s spectra were deconvoluted by fitting procedure into three peaks (Figure 4b). The peak at ca. 284.8 eV is corresponded to C-C/C-H bonds, the peak at ca. 286.0 eV is assigned with C-O/C-OH bonds, and the peak at ca. 289.2 eV may be related with O=C-O bonds. Relative portion of each peak (chemical bond) is given in Table 1.

The oxygen O 1s spectra were also deconvoluted by fitting procedure into three peaks (Figure 4c). A peak at ca. 531.8 eV is probably corresponded to some oxides or OH/O=C bonds, a peak at ca. 532.7 eV is assigned with the adsorbed H_2_O, and a peak at ca. 533.6 eV is probably due to the C-O. Other assignments of these peaks are also possible [40,41]. Relative portion of each peak (chemical bond) is given in Table 1 [42].

Table 2 gives the elemental composition of neat PU and PHL-CQDs/PU composite films samples. Based on data presented in Table 2, we observed that the content of detected elements carbon, oxygen and nitrogen decreased after PHL-CQDs nanoparticles encapsulation in polyurethane matrix.

To determine the chemical composition of PHL-CQDs nanoparticles, neat PU, and PHL-CQDs/PU composite films, we performed FTIR measurements of all samples as well (Figure 4d). In the FTIR spectrum of PHL-CQDs nanoparticles, we detected the following peaks (Figure 4d—red curves): peak at 3194 cm^−1^ stem from O-H stretching vibrations, whereas the peaks at 2934 cm^−1^ are due to C-H stretching vibrations. The peak at 1619 cm^−1^ originated from C=C stretching vibrations, whereas the peak at 1498 cm^−1^ was due to N-O stretching vibrations. The peaks at 1420 cm^−1^ could be assigned to O-H bending vibrations, whereas the peaks at 1295, 1151, and 1004 cm^−1^ stem from C-O vibrations. The peak at 804 cm^−1^ was due to C-H bending vibrations.

As for neat PU (Figure 4d—black curve), we already detected its peaks in our previous research [27]. Analysis of FTIR spectrum of PHL-CQDs/PU composite films samples showed the following peaks (Figure 4d—green curve): the peak at 3333 cm^−1^ was due to O-H stretching vibrations, whereas the peaks at 2951, 2912, and 2851 cm^−1^ could be assigned to C-H stretching vibrations. The peaks at 1728 and 1693 cm^−1^ stem from C=O vibrations. The peaks at 1615 and 1599 cm^−1^ could be assigned to C=C stretching vibrations, whereas the peak at 1529 cm^−1^ was due to N-O stretching vibrations. The peak at 1455 cm^−1^ stems from C-H bending vibrations, whereas the peaks at 1412 and 1307 cm^−1^ originated from O-H bending vibrations. The peaks at 1221, 1164, 1138, and 1076 cm^−1^ stem from C-O vibrations, whereas the peaks at 956 and 917 cm^−1^ could be assigned to C=C bending vibrations. The peaks at 852, 817, and 765 cm^−1^ stem from C-H bending vibrations [43].

### 3.4. Optical Properties of PHL-CQDs Nanoparticles and PHL-CQDs/PU Composite Films

UV–Vis spectra of PHL-CQDs nanoparticles and PHL-CQDs/PU composite films are presented in Figure 5a,b. From Figure 5a, it can be seen that there were two peaks in the absorption spectrum of PHL-CQDs nanoparticles: the first peak at 211 nm originated from π-π* transitions of the conjugated C=C bonds, and the second peak at 334 nm was due to n-π* transitions of carbonyl groups (C=O), which were detected in the XPS and FTIR spectra earlier [44,45,46]. The existence of the aromatic π-system in the core of PHL-CQDs nanoparticles was proved by the appearance of the peak at 211 nm. In the absorption spectrum of PHL-CQDs/PU composite films (Figure 5b), we identified two peaks: a peak at 321 nm and the other at 376 nm. The n-π* transitions of carbonyl groups caused the appearance of the first peak, whereas the second peak could be assigned to other surface functional groups already identified in the XPS and FTIR spectra. The peak at 211 nm could not be observed in the absorption spectrum of composite films due to the polymer being opaque for wavelengths lower than 300 nm. The blue shift of the peak at 334 nm to 321 nm of colloidal nanoparticles after encapsulation in the polymer occurs due to their transfer from a liquid to a solid environment. Complex interactions of PHL-CQDs with polyurethane matrix result in this spectroscopic phenomenon.

In addition to UV–Vis absorption spectra, we further investigated other optical properties, such as photoluminescence of both PHL-CQDs nanoparticles and PHL-CQDs/PU composite films samples. PL spectra of PHL-CQDs nanoparticles and PHL-CQDs/PU composite films are presented in Figure 6a–c. This figure shows that there is excitation–emission dependence of PL spectra of PHL-CQDs nanoparticles—Figure 6a. The PL spectra consist of three maximums: one is at 400 nm for excitations of 350 and 375 nm, respectively. The second peak is at 425 nm for excitations of 350 and 375 nm as well, and the third peak is at 450 nm for the same excitations. Thus, there are red shifts between the excitation and emission wavelengths. The PL spectra of PHL-CQDs nanoparticles are upshifted and these nanoparticles emit blue–green light. The shifts are from 53 to 75 nm depending on the excitation wavelength. Figure 6b shows fitted PL spectra of PHL-CQDs nanoparticles under excitation of 450 nm wavelength. The PL spectrum was fitted to two Gaussian curves, P1 and P2. The P1 spectrum originates from core (hexagonal carbon network), whereas the P2 peak stems from the defects from the surface and edges of CQDs [47].

Appendix A lists data (peak position and FWHM) of fitted PL spectra of PHL-CQDs nanoparticles. Based on the data from Appendix A, we could observe the upshifts of certain emission peaks. These data indicate that at lower excitation wavelengths, the photoluminescence phenomenon originates from the core of PHL-CQDs dots, but with increasing excitation wavelength, i.e., at 500 nm excitation wavelength, the photoluminescence stems from the defects on the surface and edges of PHL-CQDs induced by sp^3^ hybridization and oxygen functional groups.

Figure 6c presents PL spectra of PHL-CQDs/PU composite films. It can be seen from this figure that there is excitation–emission dependence and the highest PL intensity was for excitation of 400 nm. All spectra are upshifted and the biggest shift of 75 nm was for excitation wavelength of 325 nm. These spectra indicate successful encapsulation of PHL-CQDs nanoparticles into polyurethane polymer matrix because composite films emit blue light.

### 3.5. Reactive Oxygen Species Production

Different types of carbon dots are well known as a good photosensitizer material [11,48]. It means that they generate reactive oxygen species (ROS) under visible light illumination. There are two possible mechanisms of ROS production: one involves energy transfer from photoactive material to molecular oxygen, and the other undertakes the electron transfer from photoactive material to molecular oxygen [49,50]. Figure 7a,b show singlet oxygen production measured by two methods: measurement of PL spectra of SOSG at 530 nm (Figure 7a) and direct measurement of luminescence of singlet oxygen at 1270 nm (Figure 7b). From Figure 7a we can observe that the intensity of PL peak increases with time (5–120 min). Figure 7b shows a small broad luminescence peak at 1270 nm which could be assigned to the production of singlet oxygen.

In our investigation, we tried to determine the generation of hydroxyl radicals by measurement of the luminescence of hydroxyterephthalic acid, which was obtained by immersion of a composite films sample in hydroxyterephthalic acid. But the measurement showed that there was no signal of hydroxyl radical production. Thus, we concluded that this type of composite film produced singlet oxygen only.

### 3.6. Antibacterial Properties of PHL-CQDs/PU Composite Films

Due to specific nano-electrical and nano-mechanical properties, it is very interesting to study antibacterial activity of these composite films and, thus, their possible usage as antibacterial surfaces in healthcare facilities. We tested the composite films on various bacteria strains (*S. aureus*, MRSA, *E. faecalis*, *P. aeruginosa*, *K. pneumonie*, *L. monocytogenes*, *E. coli*, and *A. baumanii*), i.e., most of them are present in hospitals on highly touched objects. Listed in Table 3 are the antibacterial activities of PHL-CQDs composite films against these bacteria strains.

The obtained results showed that MRSA, *E. faecalis*, *E. coli*, and *A. baumanii* are the most sensitive bacteria strains on PHL-CQDs/PU composite films. However, for *P. aeruginosa*, an increased number of viable cells is observed after exposure to blue light for 1 h.

MRSA is often found in hospitals due to long-term hospitalization, recent hospitalization, invasive procedures, recent antibiotic use, etc. [51].

The mechanism of antimicrobial action of the investigated composite films is based on the following: PHL-CQDs encapsulated in the interior of PU polymer matrix generate singlet oxygen (Figure 7) and through the pores on bacteria wall membranes, this toxic form of oxygen enters the bacteria and kills them [27].

### 3.7. Antibiofouling Properties of PHL-CQDs/PU Composite Films

Apart from testing of antibacterial activity against eight bacteria strains, we conducted investigation on antibiofouling activity of PHL-CQDs/PU composite films toward three bacteria biofilms. Bacteria biofilms exist in the form of clusters attached to a certain surface or embedded each other in polymer matrix. These matrices consist of proteins, polysaccharide, and eDNA and may be cause of many human diseases. Figure 8a–c present antibiofouling activities of PHL-CQDs/PU composite films against three bacteria biofilms: *P. aeruginosa*, *S. aureus,* and *E. coli*. Figure 8a shows that *P. aeruginosa* is not sensitive on PHL-CQDs/PU composite films, whereas PHL-CQDs/PU composite films eradicate *S. aureus* biofilm completely under blue light irradiation for 1 h (Figure 8b). *E. coli* biofilm was reduced by almost 75% after blue light irradiation for 1 h. *P. aeruginosa* is a Gram-negative bacterium. The main properties of this bacterium are its metabolic versatility and ability to colonize animal hosts, ecological niches, and water environments [52]. In human beings, a lot of serious diseases can be caused by *P. aeruginosa*. This bacterium exhibits high resistance to various antibiotics caused by low outer membrane permeability. *S. aureus* is an important pathogen because it can be found in many hospitals on different surfaces. It causes minor skin infection to serious tissue infection and sepsis. The action of *S. aureus* is based on the generation of surface proteins. These proteins cause the secretion of extracellular toxins and bacterial adherence to host tissues [53].

The bacterium *E. coli* is the main cause of many diseases, including urinary tract infections, neonatal meningitis, acute enteritis in humans, and sepsis [54]. It is rod-shaped with regular dimensions of approximately 1.5 µm long and 0.5 µm wide.

Since these composite films produce only singlet oxygen under blue light irradiation, this toxic form of oxygen is a possible pathway for bacteria eradication, i.e., it causes photodynamic inactivation of bacteria strains [55].

### 3.8. Cytotoxicity

To study the cytotoxicity of PHL-CQDs/PU composite films, we used human keratinocytes cell lines (HaCaT). All experiments were conducted in dark conditions. HaCat cells are the common type of skin cells found in epidermis and they divide in the basal to spinous layer. They play the key role in the wound healing treatment. These cells form a barrier against environmental effects such as heat, UV radiation, water loss, and pathogen microbes (bacteria and viruses). In this way, it is of a great importance to investigate if the usage of PHL-CQDs/PU composite films as antibacterial surfaces (coatings) can damage these cells [56,57]. Figure 9 presents the cytotoxicity of neat PU (control sample) and PHL-CQDs/PU composite films. From this figure, we observe that the control sample did not show toxicity in any concentration of extract. PHL-CQDs/PU composite films did not show any toxicity at concentrations of 12.5, 25, and 50% either, but these samples showed mild cytotoxicity in concentration of 100% of extract.

## 4. Discussion

In this research, we studied structural, optical, antibacterial, antibiofouling, and cytotoxic activities of PHL-CQDs/PU composite films. The PHL-CQDs nanoparticles were produced by a solvothermal procedure with phloroglucinol used as precursor, and AFM analysis of more than 20 images showed that about 90% of dots had diameters up to 5 nm. EFM microscopy showed that electron density depended on the particle’s diameter: particles with lower diameter had homogeneous electron density, whereas the particles with diameter bigger than 5 nm had inhomogeneous electron density. PHL-CQDs distribution inside the polymer matrix was almost uniform. Blue-light-triggered polymer composite films produced singlet oxygen. All these data indicate the possible antibacterial and antibiofouling action of the PHL-CQDs/PU composite films.

Until now, several investigations have been conducted to check antibacterial activity of polyurethane-based composites encapsulated by metal nanoparticles (ZnO, MgO, and Mg-ZnO), and crystal violet showed antimicrobial activity [58], whereas polyurethane impregnated with dye (methylene blue and toluidine blue) killed *S. aureus* under white illumination with great efficacy [59]. These composites encapsulated with gold nanoparticles eradicated bacteria better by 36%.

Antibacterial tests indicate that the MRSA bacterium is the most sensitive to the action of the PHL-CQDs/PU composite films, whereas *P. aeruginosa* is not sensitive at all. Furthermore, the number of viable *P. aeruginosa* bacteria after 1h blue light irradiation increases. We obtained similar results related to *P. aeruginosa* biofilms. But the *S. aureus* biofilms were eradicated completely after blue light irradiation for 1 h, whereas the biofilms of *E. coli* were eradicated to 75% under the same conditions. These composite films did not show any cytotoxicity. MRSA bacteria and *S. aureus* biofilms can be found in hospitals in many places, i.e., on highly touched objects. Thus, the usage of PHL-CQDs/PU composite films can reduce nosocomial infections related to MRSA and *S. aureus*. These composite films are less potent compared to CPDs/PU and C_60_/PU, depicted in our previous research [27], but their bactericidal effect on MRSA is good enough for their usage as antimicrobial coatings on highly touched objects in healthcare facilities.

## 5. Conclusions

In this paper, we studied how structural properties (morphology, electronic, and viscoelastic) affect antibacterial, antibiofouling, and cytotoxic properties of PHL-CQDs/PU composite films. Using AFM analysis, it was found that about 90% of nanoparticles had diameter up to 5 nm and spherical shape. Smaller PHL-CQDs nanoparticles have homogenous electron distribution and Young’s modulus, while bigger ones have inhomogeneous structure. Antibacterial and antibiofouling tests indicated that these polymer composite films were highly potent in eradication of MRSA and *S. aureus* biofilms. In this way, these composite films are promising candidates for use in hospitals and healthcare facilities, especially for highly touched objects.

## Figures and Tables

**Figure 1 polymers-16-01646-f001:**
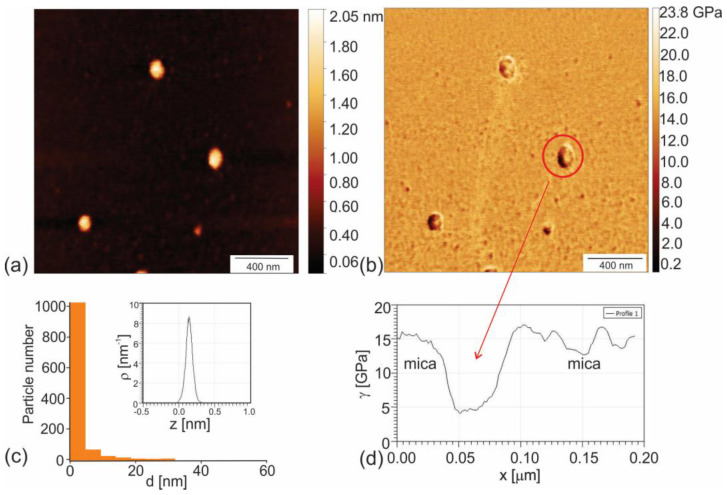
(**a**) AFM image of PHL-CQDs nanoparticles (height retrace mode); (**b**) AM-FM image of Young’s modulus of elasticity of PHL-CQDs nanoparticles (Young’s retrace); (**c**) Particle size distribution of PHL-CQDs nanoparticles and their height (figure inset); (**d**) Profile of Young’s modulus of corresponding PHL-CQDs nanoparticles designated by red arrow.

**Figure 2 polymers-16-01646-f002:**
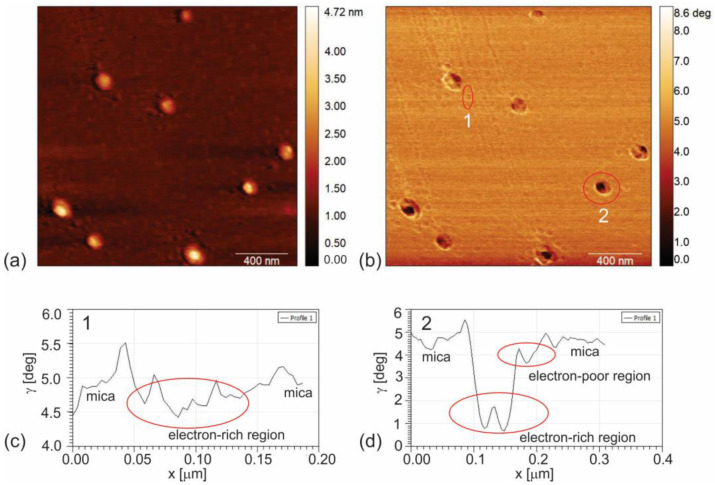
(**a**) AFM image of PHL-CQDs nanoparticles (height retrace); (**b**) EFM image of PHL-CQDs nanoparticles (nap phase retrace); (**c**) Corresponding profile of particle number 1 (representative of small nanoparticles with diameters lower than 5 nm); (**d**) Corresponding profile of particle number 2 (representative of big nanoparticles with diameters higher than 5 nm). Voltage of tip was set to 3 V, whereas ambient humidity was 35%.

**Figure 3 polymers-16-01646-f003:**
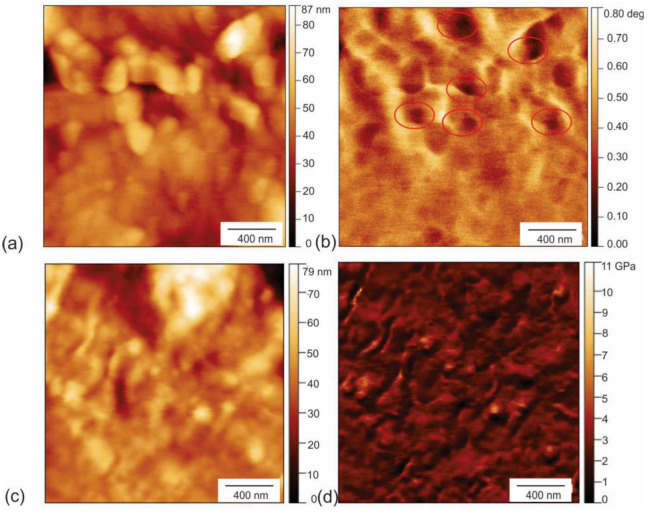
(**a**) Top view AFM image of PHL-CQDs/PU composite films sample (height retrace); (**b**) Corresponding EFM image of PHL-CQDs/PU composite films (black voids represent PHL-CQDs encapsulated into polymer matrix); (**c**) Top view AFM image of PHL-CQDs/PU composite films; (**d**) Corresponding AM-FM image of PHL-CQDs composite films. Tip voltage was 3 V, room humidity was 35%.

**Figure 4 polymers-16-01646-f004:**
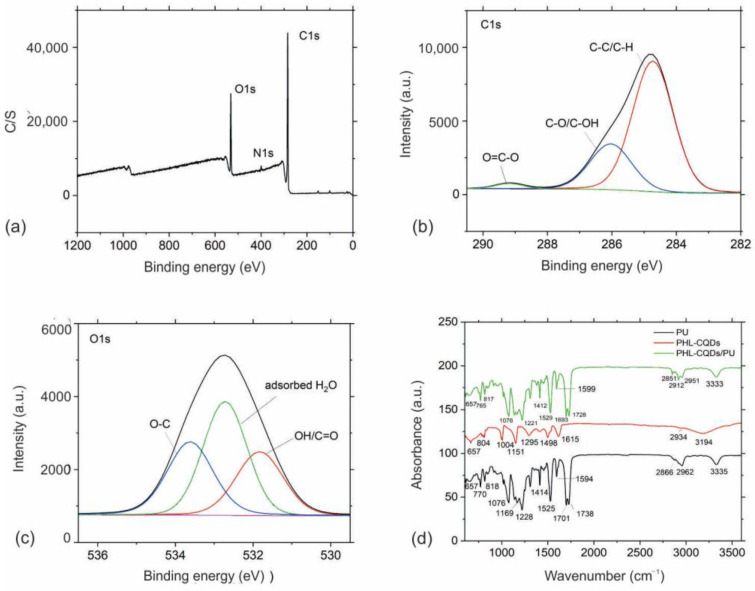
(**a**) XPS survey spectrum of PHL-CQDs nanoparticles; (**b**) High-resolution XPS spectrum of C1s peak of PHL-CQDs nanoparticles; (**c**) High-resolution XPS spectrum of O1s peak of PHL-CQDs nanoparticles; Black curve presents envelope spectra of C1s and O1s peaks; (**d**) FTIR spectra of neat PU (black curve), PHL-CQDs nanoparticles (red curve), and PHL-CQDs/PU composite films (green curve).

**Figure 5 polymers-16-01646-f005:**
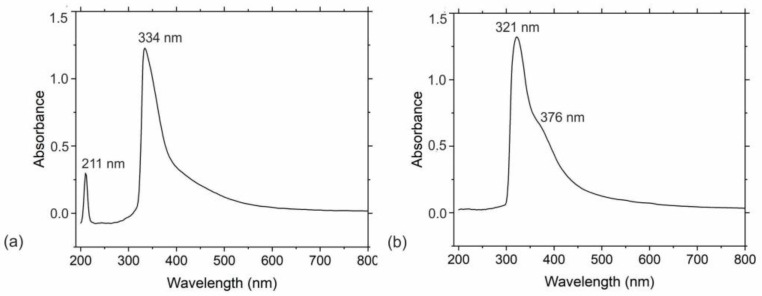
(**a**) UV–Vis spectra of PHL-CQDs nanoparticles and (**b**) PHL-CQDs/PU composite films.

**Figure 6 polymers-16-01646-f006:**
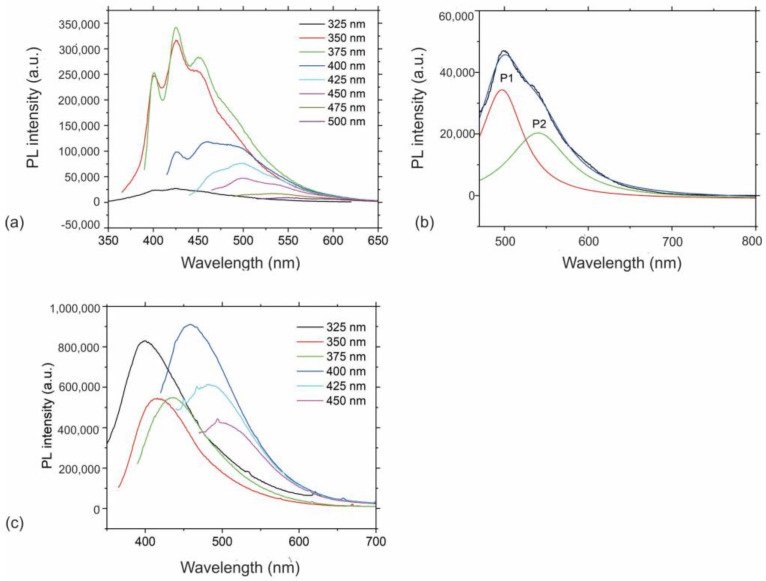
(**a**) PL spectra of PHL-CQDs nanoparticles under different excitation wavelengths; (**b**) Fitted PL spectra of PHL-CQDs nanoparticles under excitation of 450 nm wavelength. PL spectrum were fitted by two Gaussian spectra P1 and P2; P1 originated from core of PHL-CQDs (red curve); P2 peak stems from the defects from the surface and edges of CQDs (green curve); Blue curve presents fitted envelope spectrum whereas black curve presents experimental envelope spectrum; (**c**) PL spectra of PHL-CQDs/PU composite films.

**Figure 7 polymers-16-01646-f007:**
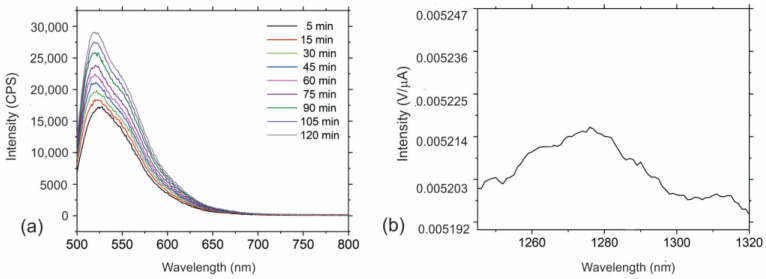
(**a**) PL spectra of SOSG used as fluorescence probe at 530 nm in the presence of PHL-CQDs/PU samples; (**b**) Intensity of luminescence of singlet oxygen at 1270 nm.

**Figure 8 polymers-16-01646-f008:**
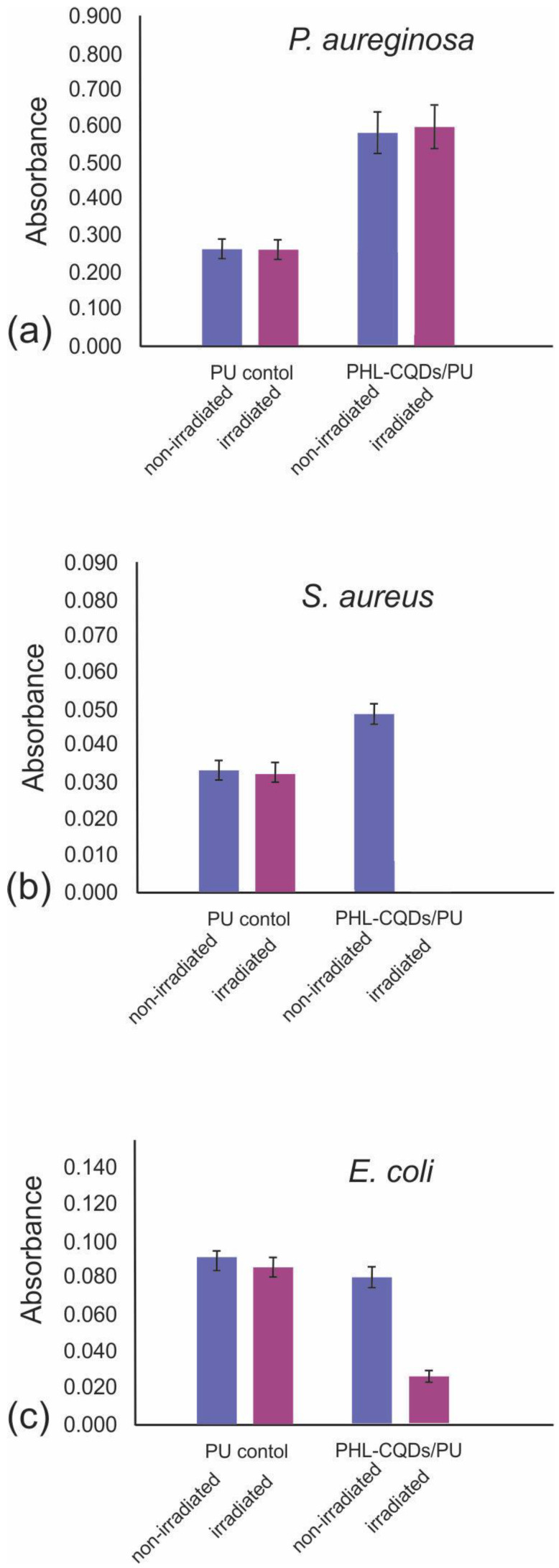
Antibiofouling effect of neat PU (control sample) and PHL-CQDs/PU composite films on *P. aeruginosa* (**a**), *S. aureus* (**b**), and *E. coli* (**c**) bacterial biofilms with and without blue light irradiation for 1 h. Absorbance is an indicator of the quantity of biofilm formed.

**Figure 9 polymers-16-01646-f009:**
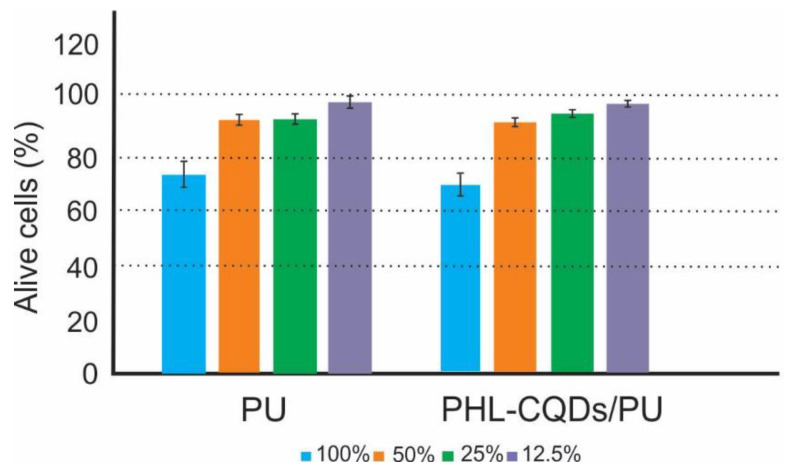
Cytotoxicity of PU (control) and PHL-CQDs/PU composite films samples without blue light irradiation determined as a percentage of viable HaCaT cells. The dashed lines highlight the limits of viability according to ISO Standard 10993-5:2009: viability > 80 corresponds to no cytotoxicity, >60–80 mild cytotoxicity, >40–60 moderate toxicity, and <40 severe toxicity [36].

**Table 1 polymers-16-01646-t001:** Elements detected in PHL-CQDs nanoparticles sample in At%; characteristic bonds identified in PHL-CQDs nanoparticles sample by XPS measurements.

Sample	C (At%)	O (At%)	N (At%)	Characteristic Bond	Binding Energy (eV)	% of Bonds
PHL-CQDs	81.9	16.8	1.3	C-C/C-H	284.8	71.1
				C-O/C-OH	286.0	25.9
				O=C-O	289.2	2.1
				OH/O=C	531.8	26.6
				adsorbed H_2_O	532.7	42.5
				O-C	533.6	30.1

**Table 2 polymers-16-01646-t002:** Elements detected in neat PU and PHL-CQDs/PU composite films samples in At%.

Sample	C (At%)	O (At%)	N (At%)
PU	91.3	6.8	1.9
PHL-CQDs/PU	90.3	6.1	3.2

**Table 3 polymers-16-01646-t003:** Antibacterial activity of the PHL-CQDs/PU composite films under blue light irradiation during 1 h.

Bacteria Strains	R
*S. aureus*	2
MRSA	7.7
*E. faecalis*	2.3
*P. aeruginosa*	0
*K. pneumonie*	1.3
*L. monocytogenes*	0
*E. coli*	5
*A. baumanii*	5.4

## Data Availability

Data are contained within the article and Appendix A.

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
