# Peer review of "Phloroglucinol-Based Carbon Quantum Dots/Polyurethane Composite Films: How Structure of Carbon Quantum Dots Affects Antibacterial and Antibiofouling Efficiency of Composite Films"

_polymers, 2024, doi:10.3390/polym16121646_

Round 1

Reviewer 1 Report

Comments and Suggestions for Authors

The article “Phloroglucinol based carbon quantum dots/polyurethane composite films: How structure of carbon quantum dots affects antibacterial and antibiofouling efficiency of composite films” reviews how structural properties (morphology, electronic and viscoelastic) affect antibacterial, antibiofouling and cytotoxic properties of Carbon quantum dots (PHL-CQDs) polyurethane (PU) (PHL-CQDs/PU) composite films. This study is promising since it was demonstrated that proposed antimicrobial surfaces allow to cut transmission path of microbes from various highly touched objects toward clinical personnel and patients.

Antibiofouling activity and toxicity of PHL-CQDs/PU composite films were investigated as well.  It was demonstrated that PHL-CQDs/PU did not show any toxicity in concentration of 12.5, 25 and 50% as well, but these samples showed mild cytotoxicity in concentration of 100 % of human keratinocytes cell extract.

The manuscript is well organised and I do recommend this article for publication in Polymers after minor revision: in the conclusions, more properties that characterize PHL-CQDs/PU should be presented by linking them to the mechanisms of antibacterial action.

Author Response

Dear Sir,

We read the referee comments carefully and revised the manuscript according to his suggestions.

Answers to referee 1:

1.The manuscript is well organized and I do recommend this article for publication in Polymers after minor revision: in the conclusions, more properties that characterize PHL-CQDs/PU should be presented by linking them to the mechanisms of antibacterial action.

We added the possible antibacterial action mechanism of these composite PU films (section 3.6, page 12).

Best regards

Biljana Todorovic Markovic

Reviewer 2 Report

Comments and Suggestions for Authors

Comments to the Author
The authors of the manuscript polymers-3021764-peer-review-v1 investigated the effects of incorporation Carbon QD on the structural and antimicrobial properties of Polyurethane. The authors mostly used AFM to visualize PHL-CQDs/PU morphology and estimate their structure and electrical characteristics. Authors used XPS and FTIR to investigate their chemical composition. Further, they examined the optical properties of synthesized PHL-CQDs films through measuring the UV-Vis and PL characteristics.  The topic is interesting and suited for the Journal scope. However, a few remarks need to be addressed before the manuscript is to be accepted.

-          Introduction…. Please e give reference for the following sentence “These dots have plenty of hydroxyl groups on their basal plane as well as carbonyl groups on the edges of honeycomb structure” Knowing that Ref. 24 gives theoretical model estimation.

-          Surface morphology of PHL-CQDs nanoparticles. It would be better if authors provided TEM images for the synthesized PHL-CQDs composite film. The AFM is not proper technique to visualize the carbon QD.

-          Please provide graphs for the XPS (survey and high-resolution C 1s, O 1s) and the FTIR for the PHL-CQDs/PU sample and plot the XPS results within the manuscript.

-          Please include the FTIR graph (Figure S3) within the manuscript.

-          Please include the UV-vis graph (Figure S4) within the manuscript, and comment on the blue-shift for the peak 334 to 321 nm.

-          Based on Fig. 7, the incorporation of PHL-CQDs has almost NO effect on the Cytotoxicity of the PU. Please explain.

Author Response

Dear Sir,

We read the referee comments carefully and revised the manuscript according to his suggestions.

Answers to referee 2  

  1. Introduction…. Please give reference for the following sentence “These dots have plenty of hydroxyl groups on their basal plane as well as carbonyl groups on the edges of honeycomb structure” Knowing that Ref. 24 gives theoretical model estimation.

We inserted ref 25 in the introduction.

  1. Surface morphology of PHL-CQDs nanoparticles. It would be better if authors provided TEM images for the synthesized PHL-CQDs composite film. The AFM is not proper technique to visualize the carbon QD.

We cannot afford cryoTEM for visualizing PHL-CQDs/PU composite films at the moment. Figure 3b represents EFM image of PHL-CQDs/PU composite films (black voids represent PHL-CQDs encapsulated into polymer matrix). The EFM technique enables the observation of charged nanoparticles inside polymer matrix.

  1. Please provide graphs for the XPS (survey and high-resolution C 1s, O 1s) and the FTIR for the PHL-CQDs/PU sample and plot the XPS results within the manuscript.

We inserted new Figure 4 in the manuscript.

  1. Please include the FTIR graph (Figure S3) within the manuscript.

We inserted FTIR spectra in the manuscript (Figure 4d).

  1. Please include the UV-vis graph (Figure S4) within the manuscript, and comment on the blue-shift for the peak 334 to 321 nm.

We inserted UV-Vis graph in the manuscript and comment the blue shift of the peak from 334 to 321 nm (Figure 5).

  1. Based on Fig. 7, the incorporation of PHL-CQDs has almost NO effect on the Cytotoxicity of the PU. Please explain.

CQDs as a new class of carbon-based nanomaterials can be synthesized from different precursors including phloroglucinol which is well known as natural product isolated from plants, algae and microorganisms. None of these CQDs did not show cytotoxicity under dark conditions (N. Stankovic et al., ACS Sustain. Chem&Eng 2018,6, 4154, M. Kovacova, ACS Biomater. Sci. Eng, 2018, 4, 3983). Because of that CQDs encapsulated in the polymer matrix (polyurethane) did not affect the cytotoxicity of PHL-CQDs/PU composite films. Cytotoxicity of materials used for photodynamic therapy is investigated under dark conditions (T. Dai, Photodiagn. Photodyn. Ther. 2009, 6, 170−188).

Best regards

Biljana Todorovic Markovic

Round 2

Reviewer 2 Report

Comments and Suggestions for Authors

I have reviewed the revised polymers-3021764-peer-review-v2 manuscript. The authors made a real effort to improve the manuscript.  They have answered my inquiries. Therefore, I recommend the last revised version for publication in Polymers